# Changes in the Ultrastructure of the Bladder Urothelium in Patients with Interstitial Cystitis after Intravesical Injections of Platelet-Rich Plasma

**DOI:** 10.3390/biomedicines10051182

**Published:** 2022-05-20

**Authors:** Yu-Khun Lee, Yuan-Hong Jiang, Jia-Fong Jhang, Han-Chen Ho, Hann-Chorng Kuo

**Affiliations:** 1Department of Urology, Hualien Tzu Chi Hospital, Tzu Chi Medical Foundation, Buddhist Tzu Chi University, Hualien 970, Taiwan; leeyukhun@gmail.com (Y.-K.L.); redeemerhd@gmail.com (Y.-H.J.); alur1984@hotmail.com (J.-F.J.); 2Department of Anatomy, Buddhist Tzu Chi University, Hualien 970, Taiwan; hcho@gms.tcu.edu.tw

**Keywords:** interstitial cystitis, bladder pain syndrome, platelet-rich plasma, electron microscope

## Abstract

Urothelial dysfunction is considered a key pathological mechanism of interstitial cystitis/bladder pain syndrome (IC/BPS). Intravesical platelet-rich plasma (PRP) injections might be effective for treating IC/BPS. This prospective study investigated the changes in electron microscopic findings among IC/BPS patients after intravesical PRP injections. Twenty-six patients with refractory non-ulcer IC/BPS underwent monthly intravesical PRP injections for 4 months. Changes in clinical symptom scores and video urodynamic study parameters were assessed from baseline to after the PRP injections. A post-treatment Global Response Assessment (GRA) score ≥ 2 was considered a successful outcome. The mean GRA score was significantly higher after 4 PRP injections than at baseline. Approximately 42% of patients experienced successful outcomes after PRP treatment. Urothelial ultrastructural defects showed no significant differences between baseline and after the PRP injections. However, patients showed variable improvements in different urothelial defects (grade improvements: urothelium cell layers, 31%; umbrella cell integrity, 42%; umbrella cell surface uroplakin plaque, 54%; tight junctions between adjacent umbrella cells, 46%; lysed organelles, 58%; inflammatory cell infiltration, 31%). Patients with successful treatment outcomes showed significant improvements in urothelial tight junction defects. Repeated intravesical PRP injections are effective for improving IC/BPS symptoms as they promote urothelial ultrastructural defect recovery.

## 1. Introduction

Interstitial cystitis/bladder pain syndrome (IC/BPS) is a chronic bladder inflammatory disease with the symptoms of chronic bladder pain, high urinary frequency, and urgency. Since the discovery of this disease, many researchers have tried to determine its pathophysiology, but it is still controversial [1]. According to previous studies, urothelial dysfunction, neurogenic inflammation, neural hyperactivity, or mast cell over-activation might play an important role in the pathophysiology of IC/BPS [2]. In the investigation of the urothelial ultrastructure in cases of IC/BPS, defects of the umbrella cells and junctional complexes were identified in the bladder [3,4]. Chronic inflammation results in an increase in urothelial cell apoptosis and a decrease in cell proliferation, with subsequent poor urothelial barrier function [5]. The regenerative ability of the defective urothelium is crucial for the recovery of urothelial barrier function with any kind of treatment.

The conventional treatment modalities for IC/BPS include cystoscopic hydrodistention, intravesical hyaluronic acid instillation, intravesical botulinum toxin A injection, oral pentosan polysulphate administration, chondroitin sulphate administration, non-steroid anti-inflammatory drug administration, sacral neuromodulation, and psychotherapy [2]. However, no therapy provides a durable treatment outcome, and most treatment modalities are not backed by scientific evidence from clinical trials.

Platelets are small anucleate cells in the circulation, and they play very important roles in hemostasis, thrombosis, and wound healing. Platelets modulate the wound healing process, including activating inflammation and tissue regeneration, through the release of growth factors, cytokines, and extracellular matrix modulators [6]. Several growth factors in platelet-rich plasma (PRP), such as platelet-derived growth factor, epidermal growth factor, and transforming growth factor, facilitate epithelial proliferation, differentiation, and wound healing [6]. Some factors released by platelets and stem cells within PRP activate and enhance a new inflammation and promote the resolution of a previously unsolved inflammation, followed by axon regeneration and target reinnervation, hence eliminating neuropathic pain [7].

In our previous studies, we found that repeated intravesical injections of autologous PRP relieved clinical symptoms, with concomitant changes in related urine biomarker levels among patients with refractory IC/BPS [8,9,10]. We also demonstrated that repeated intravesical PRP injections decreased recurrent urinary tract infection episodes and potentially promoted the recovery of urothelial defects [11,12]. The rationale of these studies was based on the fundamental therapeutic effects of PRP on bladder mucosal healing, urothelial cell regeneration, and chronic inflammation elimination. Therefore, it is rational to investigate the ultrastructural changes in the urothelium in IC/BPS bladders after repeated PRP injections. This study aimed to investigate the changes in the electron microscopic (EM) findings of the urothelium in IC/BPS patients after intravesical PRP injections. The results of this study may provide robust evidence to prove the therapeutic efficacy of PRP in patients with IC/BPS refractory to conventional treatment.

## 2. Materials and Methods

We prospectively enrolled 26 patients with IC/BPS in this study. All patients had been diagnosed as having IC/BPS according to the European Society for the Study of Interstitial Cystitis criteria in 2008 [1]. Cystoscopy was performed to exclude patients with Hunner’s lesion. All patients had been managed with conventional treatment modalities, including lifestyle modification, non-steroid anti-inflammatory drugs, cystoscopic hydrodistention, intravesical instillation of hyaluronic acid, or botulinum toxin A injection for at least 6 months, but their symptoms persisted or relapsed. The present study was approved by the institutional review board and ethics committee of Buddhist Tzu Chi Genera Hospital (IRB number 108-21-A). Informed consent was obtained from all patients before enrollment in the study.

All patients received a comprehensive medical interview and physical examination. Patients were requested to keep a 3-day voiding diary prior to admission to record the functional bladder capacity (FBC), as well as the urinary frequency and number of nocturia episodes. A 10-point Visual Analog Scale (VAS) was used to evaluate the severity of bladder pain. The O’Leary–Sant symptom score (OSS) questionnaire was used to evaluate patients’ symptoms and problems related to IC/BPS. All patients had been investigated with a video urodynamic study (VUDS) to confirm the diagnosis of IC/BPS and exclude other bladder conditions, such as bladder outlet obstruction, detrusor overactivity, and neurogenic bladder dysfunction. The symptoms and VUDS parameters were recorded before the 1st PRP injection (baseline) and after the 4th PRP injection (post-treatment). The treatment outcome was assessed using the Global Response Assessment (GRA) at each time point. The GRA is a 7-point symmetric scale that captures the patient’s general response to the treatment as follows: markedly worse, −3; moderately worse, −2; slightly worse, −1; no change, 0; slightly improved, +1; moderately improved, +2; and markedly improved, +3 [13]. The treatment was considered successful if patients reported an improvement in the GRA score by ≥2.

A total of 10 mL of PRP solution, with a mean 2.5-fold platelet concentration according to the peripheral blood count, was extracted from 50 mL of the patient’s own whole blood and was prepared with the same protocol as mentioned in our previous study [8]. Eligible patients were admitted for monthly intravesical PRP injections for 4 months. The PRP was injected 1 mm into the suburothelium at the posterior and lateral walls of the bladder at 20 sites, using a 23-gauge needle and rigid cystoscopic injection instrument (22 Fr, Richard Wolf, and Knittlingen, Germany). Cystoscopic hydrodistention was per-formed to reach a bladder capacity of 500 mL for activation of the injected platelet within 10 min. Three cold-cup bladder biopsies were performed immediately after cysto-scopic hydrodistention at the first PRP injection (baseline) and after the fourth PRP injection (post-treatment observation), followed by adequate electrocauterized hemostasis. The specimens were prepared for EM investigation, and the defects of the urothelium were graded using a 4-point scale [3].

### 2.1. Transmission Electron Microscopy

The bladder biopsy specimens were immediately washed three times in cold buffer and prefixed with 2.5% glutaraldehyde/0.1 M cacodylate buffer (pH 7.3) at 4 °C for at least 1 h, followed by post-fixation with 1% osmium tetroxide/0.1 M cacodylate buffer for 1 h at room temperature. After staining with 2% aqueous uranyl acetate, specimens were dehydrated and embedded in Spurr’s resin. Ultrathin sections of 70–80 nm were cut on a Leica Ultracut R ultramicrotome, collected on formvar-coated single-slot grids, and examined under a Hitachi H-7500 transmission electron microscope (Hitachi, Tokyo, Japan) at 80 kV. The urothelium cell layer numbers, umbrella cell integrity, and anchoring junctions were investigated and graded using a 4-point scale (0, normal; 1, mild defect; 2, moderate defect; 3, severe defect) for IC/BPS bladders.

### 2.2. Scanning Electron Microscopy

The bladder biopsy specimens were prepared and fixed with glutaraldehyde and osmium tetroxide as described above. The specimens were then dehydrated through a graded series of ethanol till 100% ethanol, and the solution was replaced with 100% acetone. The specimens were critical-point dried and sputter coated with gold. They were then examined under a Hitachi S-4700 field emission scanning electron microscope at 15 kV. In both groups, the umbrella cell size and microplicae of the cell membrane were observed, and a 4-point scale was used for grading. All EM findings were graded by a single investigator who was blinded to the clinical results.

### 2.3. Statistical Analysis

Differences in quantitative symptoms and VUDS parameters between baseline and the point after PRP injections were compared using the independent *t*-test and paired *t*-test. In the patients with IC/BPS, the grading results for the EM findings at baseline and after the fourth PRP injection were analyzed using the McNemar–Bowker test. A *p* value < 0.05 was considered significant. All analyses were performed using SPSS for Windows, version 16.0 (SPSS Inc., Chicago, IL, USA).

## 3. Results

The study enrolled 26 patients with IC/BPS (mean age, 58.6 ± 14.2 years; age range, 26–86 years). No patient with Hunner’s lesion was included in this study. Table 1 shows the clinical symptom scores and VUDS parameters of the patients at baseline and after PRP injections. A significant improvement in the GRA score was noted after PRP injections compared with the score at baseline (1.35 ± 1.06 vs. 0, *p* < 0.001). There were also improvements in the OSS, VAS score, frequency, and nocturia, but these improvements were not statistically significant. Of the 26 patients, 11 (42.3%) had a successful outcome and 15 (57.7%) had an unsuccessful outcome.

Figure 1 shows the changes in urothelial ultrastructural defects before and after PRP injections, including defects in the urothelium cell layers, umbrella cell integrity, umbrella cell surface uroplakin plaque, tight junctions between adjacent umbrella cells, lysed organelles, and inflammatory cell infiltration. The ultrastructural defects were compared in the same bladder before and after PRP treatment.

Figure 2 shows the comparison of EM findings in IC/BPS patients at baseline and after the fourth PRP injection. The McNemar–Bowker test showed no significant difference in the findings between baseline and after PRP injections. After PRP injections, improvements in urothelial ultrastructural defects were noted in 31–58% of patients, whereas 23–62% of patients had stationary results and 8–23% of patients developed worsened bladder defects (Figure 3). However, the improvement rate of urothelial ultrastructural defects was significantly higher in patients with successful treatment outcomes, especially in terms of the improvement in urothelial tight junction defects (*p* = 0.025) (Figure 4).

## 4. Discussion

In this study, we compared urothelial ultrastructural defects in IC/BPS patients before and after intravesical PRP injections. The results showed variable improvements in different defects of the urothelial ultrastructure after intravesical PRP injections, but the change was only significant in terms of the improvement in the urothelial tight junction. Approximately 42% of the patients experienced a successful outcome (GRA score ≥ 2), and the overall mean GRA score was 1.35 after 4 PRP injections. Our previous study revealed that IC/BPS symptoms improve over time up to 3 months after PRP injections [10]. These results suggest that urothelial recovery after repeated PRP injections may relieve IC/BPS clinical symptoms.

The normal bladder urothelium consists of 3–6 cell layers above the basement membrane with an intact umbrella cell layer and tight junctions [3]. Previous EM studies showed abnormal junctional complexes, epithelial pleomorphism, microvilli, and mast cell activation in IC/BPS bladders [14,15,16]. Urothelial defects are more obvious in Hunner’s type IC than in non-Hunner’s type IC, owing to severe inflammation and urothelial destruction, which was also related with more severe clinical symptoms [4]. In IC/BPS bladders, the loss of uroplakin plaques in apical urothelial plaques, especially UP-III, is known to increase the permeability of the urothelium to water and urea [17,18]. Chronic inflammation is one of the major pathological mechanisms of IC/BPS [2]. Infiltration of inflammatory cells (lymphocytes and mast cells) is common in the IC/BPS urothelium. Mast cells are implicated in systemic disorders with afferent hypersensitivity and neurogenic inflammation [19]. In a recent study of recurrent urinary tract infections, we demonstrated that repeated PRP injections promoted the recovery of bladder ultrastructural defects and reduced recurrence episodes of infection [12]. In the current study, we also noted improvements in the bladder ultrastructure in 31%–58% of IC/BPS bladders after four PRP injections.

Urothelial apical cells and tight junctions between urothelial cells maintain the bladder permeability barrier function [20]. Loss of tight junctions between urothelial cells is one of the main EM findings in IC/BPS bladders [4]. Tight junction defects lead to the loss of barrier integrity and leakage of urine into the suburothelial layer, with subsequent sensory receptor activation, resulting in irritative bladder symptoms and reduced FBC [21]. IC/BPS bladders lack the regenerative ability of urothelial cells having healthy and “tight” tight junctions, owing to underlying chronic inflammation [3]. Intravesical PRP injections have been shown to improve urinary frequency, urgency, and bladder pain in IC/BPS patients in previous studies [8,9]. A positive potassium test at baseline turned negative after repeated PRP injections [8]. The potassium test is used to assess urothelial permeability; therefore, a negative potassium test might indicate recovery of a urothelial barrier defect [22,23]. An immunohistochemistry study also showed that PRP injections increased the expressions of urothelial barrier function proteins (E-cadherin and ZO-1) and cell proliferation proteins (sonic hedgehog protein) in IC/BPS patients [24]. In the current study, we found that patients with successful outcomes had a significantly higher rate of recovery of tight junctions after PRP injections.

Currently, there is no standard preparation for PRP, especially for bladder injection [25]. In PRP, the plasma may contain anti-platelet factors that might inhibit platelet activation. A recent animal study revealed that adding saline instead of plasma to the platelet pellet could improve the therapeutic effects on would healing and angiogenesis [26]. Moreover, the platelet count in the PRP solution influences its therapeutic efficacy, and several studies demonstrated that a platelet count of 5- to 7.5-times the peripheral platelet count shows the best therapeutic efficacy [25]. Nevertheless, a recent study comparing the therapeutic efficacy of high-dose PRP (10 mL PRP from 100 mL of whole blood) with that of low-dose PRP (10 mL PRP from 50 mL of whole blood) demonstrated that the therapeutic duration was longer with 4 low-dose PRP injections than with a single high-dose PRP injection [27]. The authors explained that the results could be attributed to increased bladder area coverage, following more PRP injections every month, which provided more durable therapeutic effects.

The main limitation of this study is the lack of a placebo control group. The hydrodistention during PRP injections might have had a minor effect on the treatment out-come. However, it is difficult to conduct a placebo-controlled study for IC/BPS patients with an invasive procedure, owing to ethical concerns. The small sample size of this study was also a limitation, and it might have reduced the power of the study and increased the margin of error. Intravesical injections and biopsy during the procedure might have caused bladder injury and led to bias in EM investigations. In addition, the follow-up time might have been too short to observe changes in the urothelial histological structure. The EM findings lack a quantitative analysis for the grading of urothelial cell defects. Moreover, the specimens were graded by a single investigator, which may have led to subjective bias in the interpretation. Further randomized controlled studies with an objective quantitative analysis and immunochemical staining for electron microscopy are needed to confirm the ultrastructural changes after intravesical PRP injections in IC/BPS patients.

## 5. Conclusions

The treatment approach of repeated intravesical PRP injections is effective for improving IC/BPS symptoms. EM findings showed significant improvements in tight junction defects after the PRP treatment in patients with a successful outcome. Repeated intravesical PRP injections have the potential to promote the recovery of urothelial ultrastructural defects in some IC/BPS patients.

## Figures and Tables

**Figure 1 biomedicines-10-01182-f001:**
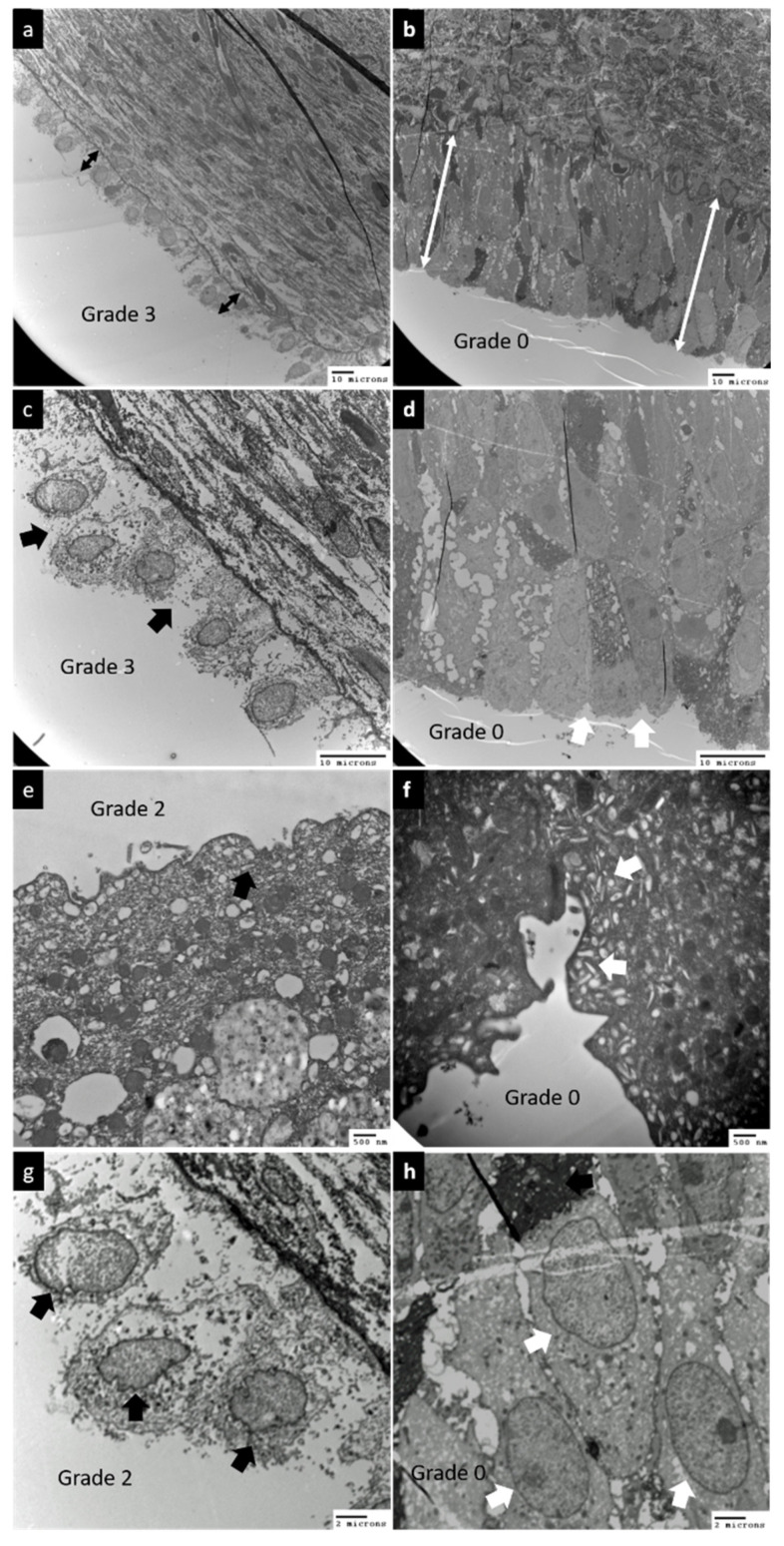
Comparison of the changes in urothelial ultrastructural defects before and after platelet-rich plasma (PRP) injections (samples from the same patient). (**a**) Before PRP: A sample with a grade 3 severe urothelial cell layer defect with only 1 urothelial cell layer (black double-headed arrow). (**b**) After PRP: A sample with grade 0 and 3–6 urothelial cell layers (white double-headed arrow). (**c**) Before PRP: A sample with grade 3 tight junction defects between >50% of umbrella cells (black arrows). (**d**) After PRP: A sample with grade 0 intact tight junctions (white arrows). (**e**) Before PRP: A sample with grade 2 decrease in uroplakin plaques (white arrows). (**f**) After PRP: A sample with grade 0 numerous uroplakin plaques (black arrows). (**g**) Before PRP: A sample with grade 2 swollen nuclei (black arrows) and ruptured organelles (black arrows). (**h**) After PRP: A sample with grade 0 normal nuclei and organelles (white arrows).

**Figure 2 biomedicines-10-01182-f002:**
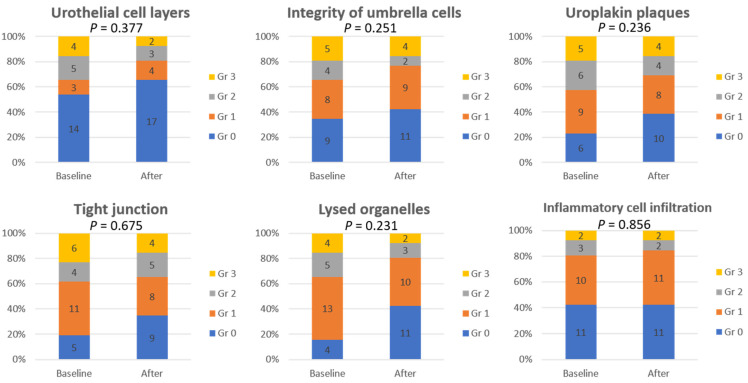
Comparison of urothelial ultrastructural defects in interstitial cystitis/bladder pain syndrome bladders at baseline and after the 4th platelet-rich plasma injection. Gr: grade of urothelial defects.

**Figure 3 biomedicines-10-01182-f003:**
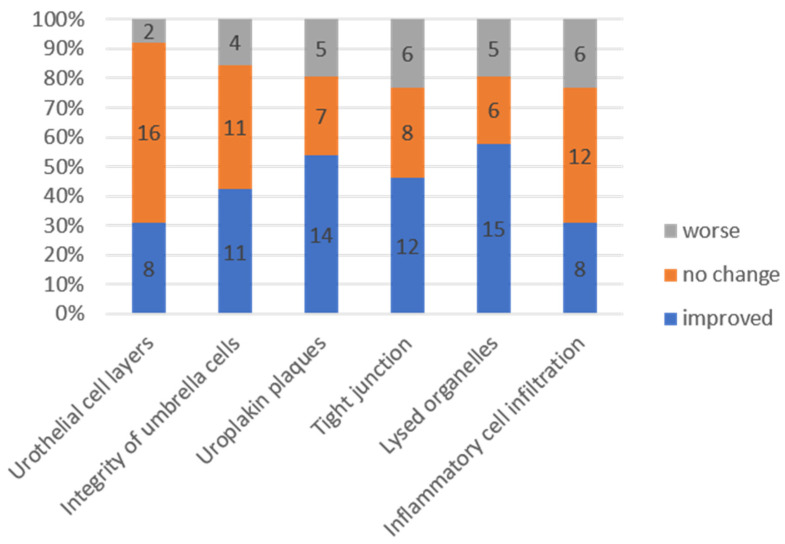
Changes in urothelial ultrastructural defects from baseline to after the platelet-rich plasma injections.

**Figure 4 biomedicines-10-01182-f004:**
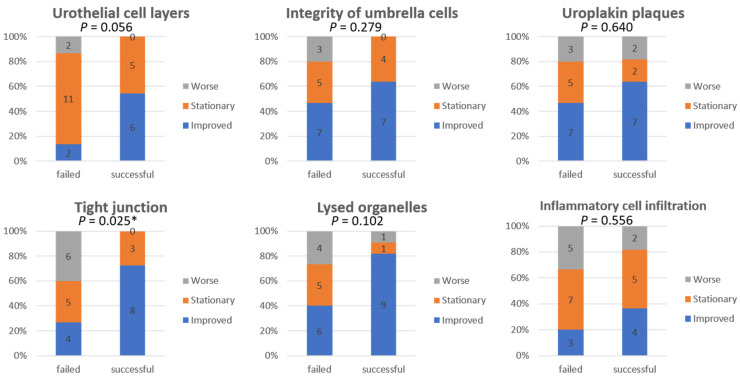
Changes in ultrastructural urothelial defects after the 4th platelet-rich plasma injection between interstitial cystitis/bladder pain syndrome patients with different treatment outcomes (failed vs. successful). * Significant difference, *p* < 0.05.

**Table 1 biomedicines-10-01182-t001:** Changes in clinical symptom scores and video urodynamic study parameters from baseline to after the platelet-rich plasma injections.

	Baseline (*n* = 26)	After the 4th PRP Injection (*n* = 26)	*p* Value
ICSI	11.60 ± 4.22	8.20 ± 3.16	0.084
ICPI	10.70 ± 3.37	8.70 ± 3.77	0.165
OSS	22.30 ± 6.72	17.30 ± 5.64	0.103
VAS	4.64 ± 3.59	2.86 ± 2.54	0.078
FBC	202.2 ± 129.5	261.2 ± 156.4	0.112
Frequency	21.40 ± 13.35	14.47 ± 9.99	0.115
Nocturia	4.17 ± 3.03	2.90 ± 1.91	0.237
Qmax	9.83 ± 5.45	10.21 ± 5.28	0.322
Voided volume	210.8 ± 116.2	225.3 ± 117.9	0.183
PVR	43.00 ± 92.39	43.54 ± 92.39	0.790
CBC	253.8 ± 103.7	268.8 ± 101.2	0.221
GRA	0	1.35 ± 1.06	<0.001 *

ICSI: Interstitial Cystitis Symptom Index, ICPI: Interstitial Cystitis Problem Index, OSS: O’Leary–Sant symptom score, VAS: Visual Analog Scale, FBC: functional bladder capacity, Qmax: maximum flow rate, PVR: post-void residual, CBC: cystometric bladder capacity, GRA: Global Response Assessment, PRP: platelet-rich plasma. * Significant difference, *p* < 0.05.

## Data Availability

Data will be made available on contacting the corresponding author.

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
