# Peer review of "Changes in the Ultrastructure of the Bladder Urothelium in Patients with Interstitial Cystitis after Intravesical Injections of Platelet-Rich Plasma"

_biomedicines, 2022, doi:10.3390/biomedicines10051182_

Round 1

Reviewer 1 Report

Congratulations, it seems to me an interesting work on the use of repeated intravesical PRP injections to improve IC/BPS symptoms.
I believe that the work is of clinical utility given the high prevalence of HF/BPS in the population

Author Response

Thank you, dear reviewer!

Reviewer 2 Report

I have a few questions for authors:
1. why biopsy after hydrodisthesion? We can expect that hydrodistrension
can change the ultrastructure of bladder mucosa, furthermore since hydrodistension is a type of treatment of IC/BPS how
can we be sure that changes are related to PRP treatment and not to hydrodistension itself?
2.  
how long was the PRP injection - the content was in the bladder (patients still excrete the drug and urine after some time, how long was that time
3.
How much time has passed from hydrodystension / biopsy to the beginning of therapy

Author Response

Dear reviewer,

Thank you for giving us the opportunity to revise the manuscript entitled “Changes in the ultrastructure of the bladder urothelium in patients with interstitial cystitis after intravesical injections of platelet-rich plasma”. Here is a point-by-point response to your comments and concerns.

Point 1:

Why biopsy after hydrodistension? We can expect that hydrodistention can change the ultrastructure of bladder mucosa, furthermore since hydrodistention is a type of treatment of IC/BPS how can we be sure that changes are related to PRP treatment and not to hydrodistension itself?

Response 1:

Before bladder biopsy, we performed hydrodistention first for better inspection of the bladder mucosa and measure maximal bladder capacity. If biopsy performed first, mucosal bleeding will interfere our procedure.

The changes of bladder mucosal ultrastructure after hydrodistention had been reported in our previous study. (Lee, Y.K., Jhang, J.F.; Jiang, Y.H.; Hsu, Y.H.; Ho, H.C.; Kuo, H.C. Difference in electron microscopic findings among interstitial cystitis/bladder pain syndrome with distinct clinical and cystoscopic characteristics. Sci Rep 2021, 11, 17258.) Cystoscopic hydrodistention was considered one of the treatment modalities for IC/BPS. However, the therapeutic effect with cystoscopic hydrodistention alone is limited, especially for the patients with refractory IC/BPS who had received conventional treatments for at least 6 months in this current study. Therefore, we considered the effect of cystoscopic hydrodistention to bladder ultrastructural change was limited. Undoubtedly, the best way is to conduct a placebo-controlled study, however, it is difficult owing to ethical concerns. (revised in limitations, page 8, line 239-240)

Point 2:

How long was the PRP injection - the content was in the bladder, patients still excrete the drug and urine after some time, how long was that time?

Response 2:

The PRP was injected about 1 mm into the suburothelium at the posterior and lateral walls of the bladder at 20 sites, using a 23-gauge needle and rigid cystoscopic injection instrument. (revised in Methods, page 3, line 98-101)

The PRP usually retained in the suburothelial space, being activated and promoted wound healing process. The regenerative process usually takes 1 month to complete, and the therapeutic effect will decline. However, currently there was no study could tell how long or how much PRP would lasting in suburothelial tissue after the treatment. Therefore, we treat IC/BPS patients with PRP injection every one month for four times. Our previous study revealed that IC/BPS symptoms may be improved over time up to 3 months after the PRP injections. (page 7, line 191-192)

Point 3:

How much time has passed from hydrodistension / biopsy to the beginning of therapy?

Response 3:

The saline was instilled at the same time during intravesical PRP injection. We completed both PRP injection and hydrodistention within 10 mins. According to AUA guideline, cystoscopic hydrodistension should performed less than 10 minutes to prevent the complications of bladder rupture and necrosis. After releasing the bladder pressure, we performed bladder biopsy immediately followed by adequate electrocauterized hemostasis. (revised in method, page 3, line 101-105)